# Tracking Changes of Chinese Pre-Service Teachers’ Aerobic Fitness, Body Mass Index, and Grade Point Average Over 4-years of College

**DOI:** 10.3390/ijerph16060966

**Published:** 2019-03-18

**Authors:** Xiaofen D. Keating, Rulan Shangguan, Kunpeng Xiao, Xue Gao, Connor Sheehan, Liang Wang, Jeff Colburn, Yao Fan, Fei Wu

**Affiliations:** 1Institute of Physical Education, Minzhu University of China, Beijing 100081, China; xk93@austin.utexas.edu; 2Department of Curriculum and Instruction, The University of Texas at Austin, Austin, TX 78712, USA; rulan@utexas.edu (R.S.); jeffcolburnpe@utexas.edu (J.C.); 3Department of Physical Education, University of Science and Technology, Changchun, Jilin 130022, China; xiaokp2008@126.com; 4Lyndon Johnson School of Public Affairs, The University of Texas at Austin, Austin, TX 78712, USA; xuegao@utexas.edu; 5T. Denny Sanford School of Social and Human Development, Arizona State University, Tempe, AZ 85287, USA; connor.sheehan@utexas.edu; 6Department of Physical Education, North East Normal University, Changchun, Jilin 130024, China; wangl493@nenu.edu.cn (L.W.); fany341@nenu.edu.cn (Y.F.)

**Keywords:** health-related fitness, academic performance, future teachers, obesity

## Abstract

Although increasing attention has been given to student academic achievement, usually measured by grade point average (GPA), and fitness in children and adolescents, much fewer studies have been conducted in higher education settings, especially in China. This study investigated the longitudinal associations of aerobic fitness (AF), body mass index (BMI), and GPA in Chinese pre-service teachers at a university. A longitudinal research design was employed to track changes in AF, BMI, and GPA, for a total of 1980 students for four years. Multi-level growth models were used to examine the interactive changes of the above three variables. It was found that GPA and BMI increased each year, while AF declined dramatically at the fourth year. The three-way interaction among GPA, gender, and BMI was significant, suggesting females who were overweight/obese had lower GPA. The data from the current study suggested that AF did not impact students’ GPA. Class standing (i.e., the fourth year) and gender (i.e., females) need to be taken into consideration when designing interventions to improve student overall fitness and academic performance in Chinese pre-service teacher populations.

## 1. Introduction

Research has suggested that there is a positive relationship between aerobic fitness (AF) and academic achievement measured by the grade point average (GPA), and a negative association between aerobic fitness (AF) and body mass index (BMI), as well as GPA and BMI [1,2,3]. Due to the pressure for academic excellence assessed by the statewide standardized testing, increasing attention has been given to the relationships among GPA, AF, and BMI in children and adolescents [4,5,6,7]. This assertion is supported by the fact that studies on the topic have been found in many countries in recent years such as China [2,8], Japan [9], Spain [10], the Netherlands [11], and the US [1,12]. The prevalence of this topic provides strong evidence reiterating the importance of such a research strand. In fact, many researchers have suggested that sound AF without weight problems is one of the most effective approaches to improve students’ health and their GPA [10,12,13]. 

It is widely known that every effort is needed to improve student GPA in education settings because it is a measure of academic achievement (AA), which is commonly used to indicate accomplishments or a level of success in academics [14,15], as a result of optimal learning efforts by students. GPA has been used both within colleges and universities, and for future employment [16,17,18,19]. Furthermore, college students are required to learn a great deal of professional knowledge and skills within a limited time frame in comparison with other educational levels where general education is of concern. The confluence of new academic challenges, stress, and the newly gained personal independence along with the financial responsibilities often results in adopting unhealthy habits (i.e., poor eating habits, as well as the lack of sleep and physical activity behaviors) among college students, putting them at a higher risk level for poor health [15,20,21,22]. Given that college students are the primary work force for future society as they are being trained to be professionals, their GPA and health-related fitness (HRF), which is the combination of four components: aerobic fitness, muscular strength and endurance, flexibility, and body composition, could have profound impact on the future development of a country [23]. Therefore, more continuous efforts are needed to improve student health and GPA simultaneously in higher education environments [13,24]. 

To date, student HRF has been studied in terms of its impact on their GPA [5,6,7,25]. Among all important HRF components, AF and BMI are the two most critical ones to the overall health of an individual [6,26], even though it is premature to conclude that AF is a better or stronger measure of HRF than BMI considering that it has been well documented that all HRF components are important and cannot be replaced by each other [27]. However, previous research has indicated that AF is a stronger predictor of hypertension risk than BMI [28]. Studies have also noted that AF plays an important role in reducing all-cause mortality and many of health problems in adults [29]. In addition, among various measures in the literature [30,31], VO_2max_ is one of the internationally accepted parameters for assessing AF [26]. With respect to BMI and HRF, Azeem and Antony [32] pointed out that “obesity is a leading risk factor for premature mortality and numerous chronic health conditions that reduce the overall quality of life” (p. 309). BMI within the acceptable range is essential to maintaining sound HRF [33,34].

It is worth noting that research on AF and GPA was only recently published, and primarily conducted in children and adolescents [6,9,24,25,35]. A systematic review on AF and GPA in youth in 2017 indicated that 20 studies examined the single relation between AF and GPA, and a strong association was found between the two variables [25]. A similar positive correlation was also reported in the study done by Muntaner-Mas and colleagues in children [36]. Kamijo and colleagues also reported a positive relation between childhood AF and brain function connectivity [37]. Although compelling evidence supported that AF is a salient contributor to GPA [10,12,25], conflicting results have been found [31]. Kwak and associates (2009) found that fitness was not associated with Swedish 9th-grade girls’ AA [31]. 

As another critical component of HRF, BMI’s role in GPA has been particularly explored due to the recent dramatic increase in weight problems among youth and young adults, regardless of gender and ethnicity [38]. Again, most studies were conducted in children and youth with contradictory results [3,5,39]. Like in studies among children and adolescents, some studies have found that lower BMI was associated with higher GPA in college students [3,38,40]. More importantly, weight gain (i.e., the increase rather than the absolute value in BMI) was found to be a more sensitive variable related to GPA [40]. However, other researchers reported an insignificant relationship between BMI and GPA [22,41,42]. In addition, Morita and colleagues found that gender played an important role in obesity and GPA [9]. The authors reported that obese girls had lower GPA, regardless of socioeconomic and behavioral backgrounds while such a difference was not found in obese boys [9,38,43]. More alarmingly, a high level of body fat was found to have negative effects on brain structure, producing long-term harmful impact on cognition [44].

When both AF and BMI were examined in relation to GPA, studies have suggested that AF was a stronger predictor of GPA than BMI in children and youth [36,45,46]. Muntaner-Mas and colleagues found that fitness may mitigate the negative effects of BMI on GPA [36], supporting the above finding that AF has stronger influence on GPA than BMI. However, to the best of our knowledge, no studies on the topic have been available in the literature for college students. Thus, it is still unclear if AF and BMI would have the same effects on GPA in the college population.

Noticeably, only a few studies have been conducted on college student AF, BMI, and GPA [24,47,48]. Using qualitative interviews for data collection, Bafail and associates did not find a significant relationship between GPA and fitness among engineering students in Saudi Arabia [47]. However, the authors suggested that other factors such as high family income, and the unique social and cultural factors in Saudi Arabia might partially prevent the authors from uncovering the expected relationship. Connaughton and associates (2003) reported a similar non-significant finding [41]. On the other hand, Scott and associates found that college female students with higher AF maintained a higher GPA [24]. However, it was also reported that there was no relationship between AF and GPA [41]. While fitness was not measured, Todd and colleagues examined the use of fitness facilities and GPA and reported a positive association [49]. A study done by Keating and associates indirectly supported the positive relationship between fitness (i.e., the frequency of strength exercise) and GPA [13]. Overall, an examination of previous research indicated that it is premature to conclude if there is a significant relationship between fitness and GPA in college students due to the scarcity of studies [13]. As such, additional work is needed to better understand the relationship between fitness and GPA among college students who live independently and often encounter a tremendous amount of stress for the first time in their lives in comparison with K-12 students [47].

While it is important to understand paired relationships among AA, AF, and BMI, the changes of the three variables are of concern. Unfortunately, there are only a few longitudinal studies on GPA, AF, and BMI using samples of children and adolescents [6,10,43]. Bartee and colleagues (2018) found that AF was related to improved math performance while no significant effect was revealed for reading in elementary school students [6]. Besides finding that AF was positively related to GPA while BMI demonstrated the opposite direction [10,22,50], studies pointed out that fitness was a better GPA predictor than BMI in a sample of K-12 students in California [43], and Nebraska [45]. A GPA gap was found between those who were persistently fit and those who were persistently unfit. Suchert and colleagues longitudinally examined the relationship of AF, BMI, and academic achievement measured by grades in math and German among German adolescents and found that there was a positive correlation between high AF and AA, while the relationship between BMI and AA remains controversial [39]. With respect of tracking changes in AF, BMI, and GPA in college students, the study reported by Meckel and associates found that college students’ AF decreased while the opposite trend was found in BMI [51]. 

More importantly, London and Castrechini pointed out that a longitudinal approach could identify trends through investigating changes over time in fitness or obesity and the effects of these changes on GPA [43]. There was only a handful of studies that have tracked changes in college student AF, BMI, and GPA in general [48], and in Chinese student populations in particular [52]. Consequently, little is known about longitudinal changes in the above three variables. The current study addresses this research need by examining changes and interactions among GPA, AF, and BMI in Chinese pre-service teachers over four academic years. The results of the current study would allow us to better understand the interactions among the three variables over time, providing baseline data for improving student AA and health. Knowledge on the topic would also enrich our understanding about the effects of AF and BMI on GPA in college students, a unique young adult population that is currently understudied. Furthermore, to date, little information on the topic is available for Chinese pre-service teachers whose GPA- and HRF-related behaviors may directly influence the quality of fostering future generations because they will be the role models for their students. This gap in knowledge undermines our endeavor of improving pre-service teachers’ AA and fitness worldwide. It is hoped that information generated by the current study would shed new light on our understanding of the associations of AF, BMI, and GPA in pre-service teachers. Moreover, knowledge regarding university students’ AF, BMI changes, and GPA by sex and majors would enrich our understanding of demographic disparities. Therefore, the purpose of this study was twofold: (a) to examine the relationships between AF, BMI, and GPA; and (b) to longitudinally explore the associations between AF, BMI, and GPA over four years in college, and differences in gender and majors. The following hypotheses were developed based on previous research findings on the topic in other populations such as children and adolescents: (a1) there was a positive correlation between AF and GPA, and BMI had a negative relationship with AF and GPA; (a2) AF and GPA increased while BMI decreased each year; and (a3) there were no differences in AF, BMI, and GPA by major and gender over the four years.

## 2. Materials and Methods 

### 2.1. Research Design and Measures

A retrospective non-experimental longitudinal research design was used to examine the associations between AF, BMI, and GPA in pre-service teachers over a duration of four years from 2012 to 2016. GPA was measured as the mean score (i.e., GPA) of courses taken by students. Weighted GPA was not employed at the university, and therefore was not used in the current study. Unlike in many western countries using a 4- or 5-point scale or letter grades, the GPA scores ranges from 0 to 100 in China.

All Chinese college students are required to take the HRF tests each year. The testing items and procedures are standardized, similar to those implemented in the US and European countries [53]. AF and BMI data were extracted from the mandated yearly fitness testing result data sets collected by the university from 2012 to 2016. VO_2max_ was used to measure AF, and the cut-off values for both sexes and first two-year vs. the last two-year students were different [52]. The raw scores in VO_2max_ data were recoded into three groups (i.e., fail, pass, and outstanding) using the specific criteria for each sex set by the Chinese Department of Sports, Hygiene and Health Education [54]. BMI was regrouped into two groups underweight/acceptable weight and overweight/obese due to the small number of those who were in the category of obese (see Table 1). 

BMI was calculated using height and weight data (i.e., weight in kg/height in meters^2^), which were measured by trained physical education instructors using stadiometers and scales, respectively. The cut-off values of BMI for underweight (i.e., BMI < 18.5), acceptable weight (i.e., 18.5 < BMI < 23), overweight and obese (i.e., BMI > 23) for Chinese college students were lower than these used in the US [30,55]. Each student’s BMI was also classified into three groups (i.e., under-weight, acceptable weight, and overweight) using the Chinese cut-off values for BMI. 

Gender information was also collected by the university. Students enrolled in 2012 had a similar age to those enrolled in other universities (i.e., 18–19 years old) because of the implementation of the centralized educational system in China, meaning that students attend school at the same age and the number of non-traditional students was very small, resulting in a small variation in age. Although there are 56 minority groups in China, no minority students enrolled in the university in 2012. Thus, age and ethnicity were not examined. It is important to note that Chinese undergraduate students usually do not switch majors and/or have more than one major [56]. Thus, no participants in the current study changed majors or had multiple majors. Unfortunately, socioeconomic status data were not available and therefore not examined in the current study. 

### 2.2. Participants

No IRB approval was needed because existing data were used. All personal information was removed before any data analysis was performed. There were 2476 students enrolled at the university in 2012. All students were required to take physical education courses for the first two years and then the courses became elective during the last two years [57]. A total of 1980 cases were used for data analyses after eliminating cases with more than 50% missing values. We conducted ancillary analyses with logistic regression models predicting if the respondent was dropped due to our exclusion criteria based on their GPA, gender, major, and AF. We found that male students with higher BMI, lower GPAs were systematically more likely to be dropped due to missing values. While the data were not missing at random, however, we are reassured that this exclusion protocol likely makes our substantive results conservative. In addition, the gender distribution of our sample of 16.7% of males and 83.3% of females was a representative sample of pre-service teachers in China even though we eliminated cases with more than 50% missing values. The percentage of social sciences and natural sciences majors was 63.4% and 36.6%, respectively. More than two thirds of students’ BMI (i.e., 77.5%) were in the category of “acceptable” (see Table 1). All participants were considered as traditional students (i.e., enrolled in the university right after their high school graduation and unmarried without children). They were full time students and lived on campus without a part-time job. There were two large student cafeterias with choices of various kinds of food and a number of in- and out-door exercise facilities free of charge. 

As noted earlier, participants in the current study were pre-service teachers who would become teachers in K-12 teachers in China. This sample is unique because the healthy behaviors of participants may affect their future students’ health through the power of role modeling [58,59,60,61] Therefore, the results of our study could provide baseline data for future research on the role of teachers in K-12 students’ adoption/maintaining of health-related behaviors. Moreover, the pre-service teacher population is also a represented sample of the college student population in China because they entered colleges/universities at the same age and lived on the same campus with other majors. They were also traditional students without any part-time jobs in general. Their required coursework load (i.e., total credit hours per semester) was also the same as non-education major students. The only difference is that pre-service teachers are required to teach in schools at the fourth year, which is away from the university campus, while other majors do not have such a requirement.

### 2.3. Data Analyses

Data screening was done first followed by descriptive analyses. Then paired correlations among AF, BMI, and GPA were calculated to examine the relationships among the three variables. Changes in student GPA, AF, and BMI were tested using growth models or multi-level models. The growth models rely on person-period data, where students are nested within each year, thus the coefficients can be interpreted as the one-year effect of the variables. In that sense, the effect sizes presented are conservative given that most students in China spend four years in college. Traditionally, researchers begin with an unadjusted model to analyze the slope of time and then interaction time with predictors to analyze how different predictors influence the slope of time [46,62]. In our case, we begin predicting GPA based on time and then subsequently interact the time variable with specific predictors to test specific hypotheses. It is important to note that the growth models estimate the average change by time [62] (see Table 2).

Specifically, we first ran a simple model documenting the change in GPA by year for the entire sample. Then AF was added followed by interacting time with BMI. Finally, to examine any gender differences we stratified our models by gender and major. These models were estimated with the xtmixed command in Stata. Cut-off values of the effect size for the F statistics were measured by partial eta squared (i.e., *η^2^*). Values for small, medium, and large *η^2^* were 0.01, 0.09, and 0.25, respectively [63]. Significance was set a priori at a *p*-value of less than 0.05. All descriptive analyses were completed using SPSS 25.0 (SPSS Inc., Chicago, IL, USA) while the multi-level models were conducted using Stata 13.0. 3 (Statacorp, College Station, TX, USA).

## 3. Results

### 3.1. Relationships of AF, BMI, and GPA in Four Years

The correlation between each pair of GPA, AF, and BMI was very small, indicating a low association among the three variables. Specifically, there were slightly negative correlations between GPA and AF throughout the first three years (*r_1_* = −0.121, *p* < 0.01; *r_2_* = −0.139, *p* < 0.01; and *r_3_* = −0.111, *p* < 0.01, respectively), which suggested that those with lower AF performed better in GPA during the first three years. Noticeably, such a negative correlation was not found at the fourth year. Moreover, the correlations between AF and BMI, although very small, were significantly positive each year (*r_1_* = 0.072, *p* < 0.01; *r_2_* = 0.104, *p* < 0.01; *r_3_* = 0.134, *p* < 0.01; and *r_4_* = 0.25, *p* < 0.01, respectively), suggesting higher BMI was associated with better AF. No significant correlations were observed between GPA and BMI, however. Overall, hypothesis a1 was not supported.

### 3.2. Changes of AF, BMI, and GPA in Four Years

Students classified as overweight/obese at the fourth year consisted of 14.2% of the sample (see Figure 1). However, AF declined as time passed with the worst at the fourth year. The percentage of students who failed the AF test at the fourth year almost doubled in comparison with that at the first year. Noticeably, the percentage of students with “outstanding” AF was only 0.7% at the fourth year in comparison with 2.6% at the first year, suggesting that only a small number of students had excellent AF at the fourth year. The changes of AF, BMI, and GPA throughout the four years were illustrated in Figure 1. Thus, hypothesis a2 was partially supported given that AF decreased as time passed.

### 3.3. GPA, AF, and BMI Trajectories

Average GPA, BMI, and AF trajectories were used to understand how the three variables changed over time (see Table 3). Students began with an average GPA score of 80.96 out of 100, and this score increased on average by 1.98 points per year. This means that by year four the average GPA for the student sample was 86.6. Students’ BMI began with an average value of 22.44 and decreased on average by 0.57 per year. Conversely, students’ AF begins with 2675.12 (SD = 609.97), and decreased on average by 50.28 per year. 

To test how GPA varied by other variables, Model 4 was used to investigate the slopes of time with AF (Note: fit group was coded as 1) and time with BMI (note: overweight/obese group was coded 1), examining how the slopes of GPA trajectories varied by AF and BMI (see Table 2). For ease of interpretability, these values are presented in the untransformed GPA. We also controlled for major (i.e., social science was coded as 1), gender, and two-way interactions between these variables. Model 4 was significant, and the most aerobically fit group began with 2.35 lower GPA on average, compared to the least aerobically fit group. However, the slope for the fit group (i.e., AF) was greater than that of the least fit group with their GPA increased additional 0.79 units each year. This suggested that the grades of the fittest group caught up the average GPA score of the least fit group in between the second and third academic year. By the end of the fourth year the average GPA of the most fit group became higher than that of the least fit group. In essence, the hypothesis a3 was not supported.

The interaction between BMI and GPA trajectories was also examined, revealing significant GPA differences between BMI groups. Specifically, overweight/obese students started with a similar GPA to non-obese ones. However, the overweight/obese group’s GPA increased at a slightly faster rate (0.22 per year, *p* < 0.05), suggesting that overweight/obese students’ GPA was higher than non-obese students’ GPA at the second year, and almost a half-point higher by the end of the fourth year in college.

### 3.4. GPA, AF, and BMI by Gender and Major

#### 3.4.1. GPA, AF, and BMI by Gender

We also investigated if there were gender differences in GPA, AF, and BMI trajectories in Model 4. The regression results for female and male were presented in Models 5 and 6, respectively. Results in Model 4 indicated that females had higher first year GPA than that for men (2.03, *p* < 0.01), but females had lower slopes than that for male students (−0.86, *p* < 0.01), which suggested that females’ GPA increased slower than that for their male counterparts. Results in Model 5 suggested that females with higher AF had lower initial GPA (−2.76, *p* < 0.01), but greater slopes of GPA (0.91, *p* < 0.01). Similarly, we found that obese females started with lower GPA than non-obese females (−0.76, *p* < 0.05) but increased at a faster rate (0.297, *p* < 0.01). The comparison of results in Models 4, 5, and 6 suggested that the significant differences observed at the university level were largely driven by the significant differences in AF and BMI among females (See Models 5 and 6). In Model 6, it was found that the beginning average GPA for males was 74.39 (*p* < 0.01) with an average of 2.80 GPA increase per year. The coefficients of the interaction between year and AF groups indicated that AF was not significantly associated with differences in baseline GPA or the GPA slope for males. Similarly, there was little difference in baseline GPA (i.e., first year) between overweight/obese and acceptable weight males. 

#### 3.4.2. GPA, AF, and BMI by Major

The differences in AF, BMI, and GPA by majors were statistically significant in Models 4–6. Students in social science initially had an average GPA that was 2.13 higher than those in natural sciences (*p* < 0.01), however, their GPA increased 0.86 slower than students’ GPA in natural sciences (*p* < 0.01). Particularly, it was also found that among social science students, those who were in the most fit group had 1.11 higher GPA than those in the least fit group (*p* < 0.05). This significant difference was mainly driven by female students, because there were no male students that fell in both the social science major and most fit group (See Models 4–6 in Table 3).

### 3.5. Effect Sizes

In order to provide practical information about the magnitude of the effects, we calculated effects’ sizes for the major significant results. For any groups (e.g., social science major vs. natural science major, male vs. female, AF groups, and BMI groups) in our dataset, sample sizes were unequal across different groups. Therefore, we need to take into account the unequal sample sizes when calculating effect sizes. Equation (1) shows the methodology we employed to calculate the effect size [64,65], where x_1_ (bar)–x_2_ (bar) is the difference between group means. S_p_ is the weighted standard deviation that takes account of unbalanced sample sizes. n_1_ and n_2_ are sample sizes of either group, and s_1_ and s_2_ are standard deviations of either group.

As for GPA, AF, and BMI trajectories, the most aerobically fit group began with 2.35 lower GPA on average compared to the least aerobically fit group, and thus the corresponding effect size is 0.58. As the GPA of the most fit group increases 0.79 greater than that of the least fit group, the true difference between groups across the college career (4 years) was 3.16 and the corresponding effect size was 0.71. The overweight/obese group’s GPA increased at a slightly faster rate (0.22 per year, *p* < 0.05) and thus the overweight/obese group’s GPA would be 0.88 higher than non-obese ones at the fourth year. The corresponding effect size was 0.19.

As for GPA, AF, and BMI by gender, Model 4 showed that females had higher first year GPA than that for men (2.03, *p* < 0.01), but females had lower slopes than that for male students (−0.86, *p* < 0.01). Therefore, the corresponding effect size of the gender variable was 0.53. Taking into account the different growth rates between females’ and males’ GPA, the true effect sizes across the 4 years in college was 0.75. For female students, the effect size of being in the good AF group was 0.79 initially, and the true effect size across the college career was 0.83. As obese females’ GAP increases at a faster rate (0.297, *p* < 0.01) than non-obese females, the corresponding effect size was 0.26.

As for GPA, AF, and BMI by major, the corresponding effect size of the major variable was 0.55. Taking into account the different growth rates between social science students’ and natural science students’ GPA, the true effect sizes across the college career was 0.76. Among social science students, the corresponding effect size of being in the most fit group was 0.29.

The most popular benchmark for gauging effect sizes is Cohen’s (1988) prescription that values of 0.20, 0.50, and 0.80 are considered small, moderate, and large, respectively. Based on the above benchmark, we found that the effect sizes regarding the gender variable, the different AF groups, and the major variable were moderate to large, while the effect sizes regarding the overweight/obese group was small.
(1)$$\mathrm{cohen}’sd=\frac{\bar{x_{1}}-\bar{x_{2}}}{s_{p}},\;\mathrm{where}\;s_{p}=\sqrt{\frac{(n_{1}-1)s_{1}^{2}+(n_{2}-1)s_{2}^{2}}{n_{1}+n_{2}-2}}$$

## 4. Discussion

The important role of such research in college student wellness and GPA, and the limited body of literature clearly point to the urgent need for more studies on the topic in the future. This line of research warrants more attention of relevant professionals in higher education. The current study marks the first attempt to longitudinally examining AF, BMI, and GPA in college students. It addresses the paucity in the education and health literature by examining the interactive relationships of AF, BMI, and GPA using a sample of Chinese pre-service teachers in a higher education setting. Notably we found: (a) both AF and BMI declined while GPA was increased each year; (b) AF and GPA were negatively correlated for the first three years and a reverse trend was shown at the fourth year while the correlation between BMI and GPA was not significant each year; and (c) significant gender effect was found in GPA, BMI, and AF. Specifically, AF, BMI, and GPA did not have any significantly interactions for male students. However, females with higher AF had lower initial GPA and outperformed the ones with lower AF academically. Overweight/obese female students had lower GPA than those with acceptable weight at the first year and an opposite relationship was found at the fourth year. 

### 4.1. Changes of AF, BMI, and GPA in Four Years of College

While it is encouraging that both GPA (i.e., increased) and BMI (i.e., decreased) improved as time passed, it is alarming that student AF significantly declined at the fourth year after it kept stable for the first three years (see Figure 1). This steep drop in AF at the last year in college, which is in line with the result found by the study done by Huang and colleagues [56] using a sample of Chinese college students, is a cause for concern considering that AF plays a critical role in the overall fitness [66] and student GPA [63]. As widely known, the last year in college is the most stressful time because students need to find a job via many job interviews in different schools. More importantly, as noted earlier, pre-service teachers are required to teach students in schools at the fourth year of college, which is more time demanding than regular coursework [67,68,69]. The steep decline at the last year of college may suggest students have not been able to maintain a sound AF level when time is limited, and stress is high. Interestingly, students’ final year GPA was the highest among the four years. This may suggest that time and stress had little impact on student academic performance. Caution must be exercised, however, when interpreting such a result given that no data are available concerning the difficult level of coursework in each academic year. It may be possible that the fourth-year study (i.e., student teaching) could be much easier, though more time demanding, than that in the previous three years, resulting in a higher GPA at the last year. On the other hand, it is important to point out that physical education is required at the first two years in Chinese higher education [56]. The data from the study seem to indicate that the two-year requirement for physical education still cannot help students maintain their fitness level at the fourth year. Therefore, more physical education or physical activity intervention programs may be needed throughout the four years in college to warrant sustained influence on college student AF. 

BMI, on the other hand, decreased each year (see Figure 1), which should be considered as positive changes in the western literature [70,71]. However, the finding from our study may be a cause for concern in Chinese college students as research has pointed out that such weight loss is actually unhealthy [72]. As well documented in the literature, being underweight or overweight is not good for health [72,73,74]. Zhang and colleagues noted that “underweight will jeopardize health in the long term, which may reduce sex hormones and bone mineral density and lead to anemia, low blood pressure, fatigue, discomfort and eating disorders, including anorexia nervosa, bulimia nervosa, and binge eating disorders” (p. 2) [72]. Given that the percentage of students classified as underweight changed from 7.3% in the first year to 27.7% in the fourth year (see Table 1), our study echoed the need for better educating Chinese college students about healthy weight management as suggested in the existing literature [52,75,76]. To date, unfortunately, more attention has been given to the effects of weight gain, and excessive weight loss on AF and GPA during college years has been severely neglected given that fewer studies on the topic have been reported [72]. 

It is important to note that our study supported the result found in children and youth that AF is a stronger predictor of GPA than BMI [45] considering that only AF was significant while BMI was not in Models 4 and 5. However, caution needs to be exercised when interpreting the result because the percentage of students in the underweight group increased each year over 4 years of college in China. More research on the topic is needed in the future.

### 4.2. Changes of AF, BMI, and GPA by Gender and Major

It was unexpected that the current study did not support the hypothesis that AF would be positively associated with GPA, while BMI and GPA would negatively relate to each other for the entire group. However, a small but positively significant relationship between GPA and BMI was found when the analysis was conducted by gender. Specifically, BMI and AF did not significantly affect GPA for males within the four years. However, there was a significance for females. Overweight/obese students had lower GPA to begin with than those with acceptable weight, which is in line with the finding reported by previous studies [3,38]. Surprisingly, overweight/obese students performed better on GPA than the acceptable weight group as time passed. This finding has not been reported in the literature due to the lack of longitudinal studies. It is important to point out, however, that issues related to weight and health among Chinese college students may be different from those in western countries because research has indicated that Chinese females may be thin but not healthy [55,72]. As a result, overweight females had better GPA than the ones with acceptable weight. 

Major is a unique variable related to college students as it is the first time that students are required to take different courses based on the specific requirements of their selected major [13]. This is due to the fact that students usually are required to take the same courses and therefore major is not a factor that influences student GPA, AF, and BMI in K-12 programs in China. Some majors consist of a great deal of reading while others may require long hours of conducting experiments in a lab setting in college. Therefore, stress and the available amount of time for fitness may vary greatly by majors. Interestingly, GPA was only affected by AF and BMI in female students in social sciences. The results from the current study were aligned with what was reported by previous studies that higher AF females in social sciences had a higher GPA than those who were least fit [4,39]. Why male students did not demonstrate the same trend as their female counterparts may be related to the fact that male students had a different experience in college considering previous research has noted that female students were under more stress than male students [23,77]. 

### 4.3. Limitations

There are important limitations that are worth noting. First, while the sample size was relatively large for a longitudinal study that lasted for 4 years, the sample cannot represent its population as all participants were from one university. Second, there were more females and social science majors without any minority participants in the current study. Moreover, all participants were traditional students with no part-time jobs. As such, cautions should be exercised when interpreting the results found in the current study.

## 5. Conclusions

Using a sample of Chinese pre-service teachers, the results of the longitudinal study revealed that BMI was not significantly associated with GPA, while AF was negatively correlated to GPA for the first three years and such a relationship was not found at the fourth year. Unlike that found in the literature [7,51], both AF and BMI decreased while GPA increased over four years in college. AF was a more salient factor influencing GPA than BMI, which was in line with previous research on the topic. The significant decline in AF and BMI in female students at the last year of college is a cause for concern because they may be thin but unhealthy. This overlooked decline in BMI among females warrants more attention of professionals in education and health in the future. More studies are needed to longitudinally track changes of AF, BMI, and GPA in college students.

## Figures and Tables

**Figure 1 ijerph-16-00966-f001:**
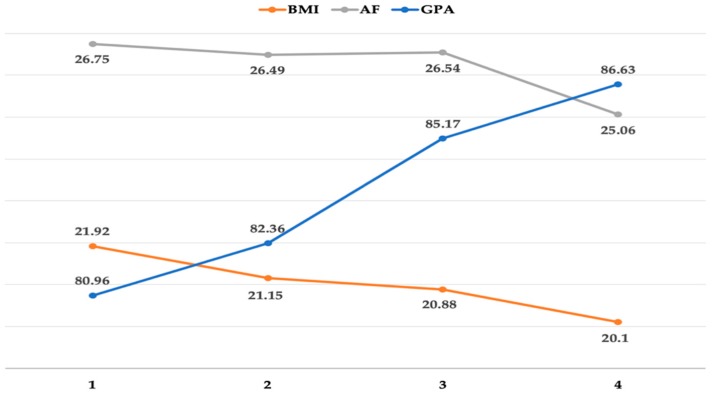
Changes of Aerobic Fitness (AF), Body Mass Index (BMI), and Grade Point Average (GPA) through the four years (Note: AF data were divided by 100 in order to fit all three variables in the same figure).

**Table 1 ijerph-16-00966-t001:** Descriptive Statistics of Grad Point Average (GPA), Aerobic Fitness (AF), Body Mass Index (BMI) by Gender, Year in College, and Major.

Variables		Freshmen	Sophomore	Junior	Senior
GPA	Female	81.23 (3.63)	82.81 (4.11)	85.58 (4.18)	86.80 (4.57)
M (SD)	Male	79.28 (4.66)	80.12 (5.27)	83.12 (4.75)	85.83 (4.77)
	Social science major	81.29 (3.63)	82.76 (4.13)	85.63 (4.14)	85.86 (4.25)
	Natural science major	80.39 (4.25)	81.67 (4.85)	84.36 (4.64)	87.97 (4.92)
	Total	80.96 (3.89)	82.36 (4.44)	85.17 (4.37)	86.63 (4.62)
AF (VO_2max_)	Female	2499.83 (417.06)	2471.91 (540.06)	2474.58 (531.92)	2341.24 (412.86)
M (SD)	Male	3551.57 (664.56)	3531.66 (660.99)	3548.39 (635.21)	3328.81 (538.92)
	Social science major	2647.50 (587.88)	2640.01 (656.79)	2656.99 (647.15)	2479.11 (555.64)
	Natural science major	2722.94 (644.04)	2722.94 (644.04)	2647.60 (735.06)	2552.10 (593.75)
	Total	2675.12 (609.97)	2648.53 (686.82)	2653.55 (680.49)	2505.84 (570.83)
% (*N*)	Fail	13.3 (263)	18.8 (373)	21.9 (433)	25.2 (499)
	Pass	74.4 (1473)	67.5 (1337)	65.6 (1298)	70.7 (1400)
	Outstanding	12.3 (244)	13.6 (270)	12.6 (249)	4.1 (81)
BMI	Female	21.94 (2.69)	21.16 (2.57)	20.83 (2.55)	19.94 (2.67)
M (SD)	Male	21.81 (2.69)	21.13 (2.52)	21.12 (2.58)	20.93 (2.89)
	Social science major	21.74 (2.71)	21.01 (2.57)	20.77 (2.54)	20.13 (2.69)
	Natural science major	22.22 (2.63)	21.40 (2.52)	21.07 (2.57)	20.05 (2.80)
	Total	21.92 (2.69)	21.15 (2.56)	20.88 (2.55)	20.10 (2.73)
% (*N*)	Underweight	7.3 (144)	13.7 (271)	16.8 (333)	27.7 (549)
	Acceptable	61.7 (1222)	64.1 (1269)	63.3 (1253)	58.0 (1149)
	Overweight	31 (614)	22.2 (440)	19.9 (394)	14.2 (282)

**Table 2 ijerph-16-00966-t002:** Academic achievement trajectory. Body Mass Index (BMI) and aerobic fitness trajectories.

Variables	Model 1	Model 2	Model 3
	GPA	Body Mass Index (BMI)	Aerobic Fitness
year	1.982 *	−0.573 **	−50.284 **
(standard error)	(0.0379)	(0.0253)	(4.369788)
Constant	80.96 **	22.44 **	2746.47 **
(standard error)	(0.0947)	(0.0633)	(10.9245)
Observations	7920	7920	7920
Individual fixed effect	yes	yes	yes

* *p* < 0.05; ** *p* < 0.01

**Table 3 ijerph-16-00966-t003:** Results of multilevel models.

Dependent Variable	Model 4	Model 5	Model 6
	GPA	GPA	GPA
	All students	Female	Male
year	2.548 ***	2.424 ***	2.799 ***
(standard error)	(0.0927)	(0.106)	(0.203)
major	2.130 ***	1.880 ***	2.865 ***
(standard error)	(0.303)	(0.327)	(0.782)
male	−2.030 ***		
(standard error)	(0.290)		
overweight & obese	−0.437	−0.762 **	1.211
(standard error)	(0.320)	(0.338)	(0.817)
1. aerobic fitness (entire sample)	reference group	reference group	reference group
2. aerobic fitness (fail)	−0.00608	−0.319	0.481
(standard error)	(0.312)	(0.341)	(0.736)
3. aerobic fitness (Pass)	0.677	0.386	0.160
(standard error)	(0.523)	(0.560)	(1.164)
4. aerobic fitness (Outstanding)	−2.349 ***	−2.764 ***	2.237
(standard error)	(0.754)	(0.747)	(3.907)
1. aerobic fitness (entire) * year	reference group	reference group	reference group
2. aerobic fitness (fail) * year	−0.0650	0.0345	−0.191
(standard error)	(0.0913)	(0.103)	(0.212)
3. aerobic fitness (pass) * year	−0.194	−0.0594	-0.472
(standard error)	(0.163)	(0.177)	(0.436)
4. aerobic fitness (outstanding) * year	0.790 ***	0.914 ***	
(standard error)	(0.260)	(0.258)	
overweight & obese * year	0.221 **	0.297 ***	−0.143
(standard error)	(0.104)	(0.114)	(0.253)
1. aerobic fitness * male	reference group		
2. aerobic fitness * male	0.0190		
(standard error)	(0.279)		
3. aerobic fitness * male	−1.163 **		
(standard error)	(0.486)		
4. aerobic fitness * male	0.335		
(standard error)	(3.474)		
overweight & obese * male	0.214		
(standard error)	(0.380)		
1. aerobic fitness * major	reference group	reference group	reference group
2. aerobic fitness * major	0.187	0.236	0.180
(standard error)	(0.228)	(0.250)	(0.567)
3. aerobic fitness * major	0.135	0.0347	0.633
(standard error)	(0.407)	(0.447)	(1.005)
4. aerobic fitness * major	1.113 **	1.214 **	0
(standard error)	(0.567)	(0.560)	(.)
overweight & obese * major	0.102	0.324	−0.901
(standard error)	(0.288)	(0.307)	(0.778)
major * year	−0.864 ***	-0.876 ***	−0.703 ***
(standard error)	(0.0690)	(0.0749)	(0.179)
Constant	77.77 ***	78.28 ***	74.39 ***
(standard error)	(0.322)	(0.352)	(0.742)
Random-effects Parameters			
var (year)	0.229 ***	0.254 ***	0.0871 *
(standard error)	(0.0190)	(0.0199)	(0.0554)
var (constant)	7.376 ***	6.536 ***	11.22 ***
(standard error)	(0.186)	(0.188)	(0.606)
var (Residual)	9.211 ***	8.699 ***	11.69 ***
(standard error)	(0.0925)	(0.0956)	(0.288)
Observation	7920	6600	1320

* *p* < 0.10; ** *p* < 0.05; *** *p* < 0.01.

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
