# Peer review of "Tracking Changes of Chinese Pre-Service Teachers’ Aerobic Fitness, Body Mass Index, and Grade Point Average Over 4-years of College"

_ijerph, 2019, doi:10.3390/ijerph16060966_

Round 1

Reviewer 1 Report

1) You must adequately organize the work reported (e.g., in chronological order, thematic, types of research, etc.). 

2)The conceptual reference must be improved and sustained with sufficient and adequate authority figures. Some updating of the references is recommended.

3)consistency between conclusions and discussion could be improved

In general, the article is of high quality and can basically be published as configured and the statistical analyses are very correct.

Author Response

Thank you very much for the invaluable suggestions and comments. We believe that we have made all the suggested changes. 

Reviewer 2 Report

This study is methodologically sound and addresses an important area, the connection between physical health and academic achievement. The main criticisms concern presentation of the results such that they are easily understandable, and generalizability.

What is the proposed mechanism of action between aerobic fitness and GPA? Is it the brain function connectivity as mentioned in line 67 or something else? Please review relevant literature or suggest a mechanism.

For aerobic fitness to be a better measure of health than BMI is an important point, especially given the study’s conclusions. Emphasize this point more in introduction and discussion.

Lines 44-58: no need to bring up link between GPA and cognition. Indeed, GPA’s usual function is as a measure of academic achievement (linked to but not identical to cognition) to be used both within the university and for future employment

Define health-related fitness (HRF) the first time it is used. Same with AA (academic achievement).

How is aerobic fitness measured in this study? What VO2 max benchmarks were used to indicate the 3 groups (fail, pass, outstanding)?

This study brings up many cross-cultural studies. Are results from this study and others expected to vary based upon ethnicity and/or geographic region?

College students are a convenient and therefore well-studied population. Are they understudied in China?

Please better justify use of this sample, beyond convenience. Are preservice teachers in some way representative of other young adults/college students?

Approximately 20% of the sample was excluded due to missing data. Was the data missing at random?

This study has a large enough sample that results may become significant due to sheer power. Therefore, display the effect sizes more prominently and interpret their practical significance.

Methodology is sound and well thought out.

The charts in Figure 1 are difficult to read. Make them larger.

If Table 2 is going to run to 2 pgs, the column labels must be carried over to both pages.

Table 2 needs to be better labeled. Are the values parameter estimates with effect sizes? If so, label them.

Line 225: whether the hypothesis was supported should be presented after the data.

The authors did well to interpret reasons for GPA possibly rising in the 4thyear. Speak more about the implications of decreasing BMI being unhealthy (loss of muscle mass, for example), especially as the sample ages further.

Is this sample generalizable: preservice teachers, entered high school directly after graduation, unmarried, no children etc? Please address this in discussion.

Also address sample imbalance (primarily female, no males who were fit and social science majors).

Author Response

Thank you very much for your invaluable comments and suggestions. We believe that we have made all the suggested changes.

Round 2

Reviewer 2 Report

the literature and background is much improved and shows a stronger rationale for the study design and selected sample.

methodology is more sound, with the examination of missing data patterns

Figure 1 is easier to read and interpert in this format

Double-check where the column headings on 2nd page of Table 3 show up. Should be on top of the page 

While the equation in lines 354 may not be strictly necessary, as effect size calculation is common knowledge, it is helpful to have it for easy reference. Effect size and their interpertation strengthen study results.

The paragraph on representativeness may be moved to description of study participants in methods section. Check spelling of "research" on line 374.

Overall this paper is much more methodologically sound, easier to interpert, and better fits with the overall literature. I recommend publication after correction of a few minor typos 

Author Response

Thank you very much for your comments and suggestions. We have made the suggested revisions. Many thanks, again.
